# Customize Your Visual Autoregressive Recipe with Set Autoregressive Modeling

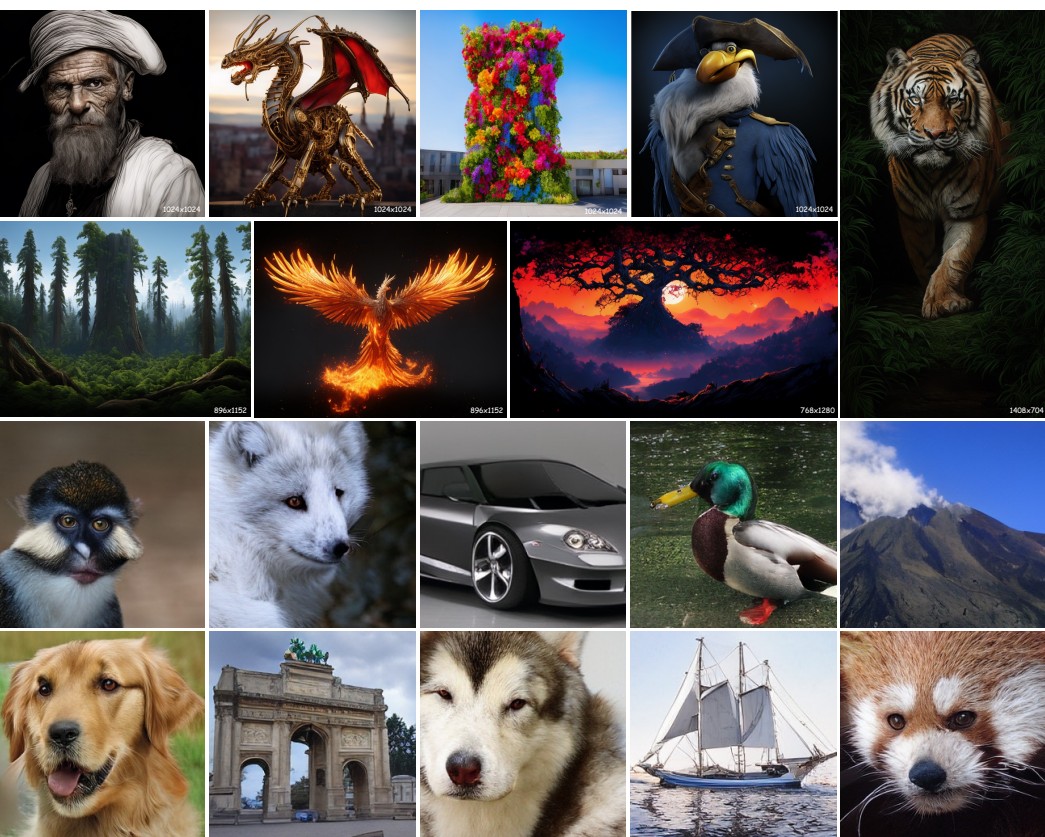

Figure 1: Text-conditioned and class-conditioned samples generated by SAR models.

## Abstract

We introduce a new paradigm for AutoRegressive (AR) image generation, termed *Set AutoRegressive Modeling* (SAR). SAR generalizes the conventional AR to the next-set setting, *i.e.*, splitting the sequence into arbitrary sets containing multiple tokens, rather than outputting each token in a fixed raster order. To accommodate SAR, we develop a straightforward architecture termed *Fully Masked Transformer*. We reveal that existing AR variants correspond to specific design choices of sequence order and output intervals within the SAR framework, with AR and Masked AR (MAR) as two extreme instances. Notably, SAR facilitates a seamless transition from AR to MAR, where intermediate states allow for training a causal model that benefits from both few-step inference and KV cache acceleration, thus leveraging the advantages of both AR and MAR. On the ImageNet benchmark, we carefully explore the properties of SAR by analyzing the impact of sequence order and output intervals on performance, as well as the generalization ability regarding inference order and steps. We further validate the potential of SAR by training a 900M text-to-image model capable of synthesizing photo-realistic images with any resolution. We hope our work may inspire more exploration and application of AR-based modeling across diverse modalities. Code will be available.

# 1 INTRODUCTION

The success of AutoRegressive (AR) models in Large Language Models (LLMs) (Radford, 2018; Radford et al., 2019; Brown, 2020; Raffel et al., 2020; Yang, 2019; Touvron et al., 2023) has also driven their development in image generation, where some recent work (Ramesh et al., 2021; Yu et al., 2021; 2022; Tian et al., 2024; Li et al., 2024; Sun et al., 2024; Liu et al., 2024) has demonstrated that the generative capabilities of AR models can rival or even surpass those of diffusion models (Song & Ermon, 2019; Song et al., 2020b; Ho et al., 2020; Dhariwal & Nichol, 2021; Rombach et al., 2022; Song et al., 2020a; Lipman et al., 2022; Liu et al., 2022; Karras et al., 2022; Peebles & Xie, 2023; Esser et al., 2024; Gao et al., 2024; Zhuo et al., 2024).

Despite their strong performance, the large number of inference steps in AR models due to the 'next-token prediction' manner has become a bottleneck. This limitation has inspired explorations on more efficient AR approaches, with the idea of outputting multiple tokens simultaneously. Existing work (Chang et al., 2022; Yu et al., 2023a; Chang et al., 2023; Li et al., 2023; 2024; Ni et al., 2024) usually adopts BERT-like (Devlin, 2018) masked modeling approaches to exchange the cost of always performing global computations (thus KV cache is not allowed) for fewer inference steps. Another stream of work designs proper sequence orders and arranges multiple tokens with similar properties into one group, to predict these tokens at once, *e.g.*, the scale-aware order (Tian et al., 2024; Zhang et al., 2024; Ma et al., 2024). We conclude that, in the training phase, these approaches pay attention to two aspects: one is the *sequence order*, the other is the *output intervals*. The defined order and intervals split the sequence into token sets. AR splits the sequence into sets of single tokens, VAR (Tian et al., 2024) builds several multi-scale sets for an image, and Masked AR (MAR) (Chang et al., 2022; Li et al., 2023; 2024) randomly divides the sequence into a masked set and an unmasked set. Fig. 2 (a1, a2) illustrates examples for AR with intervals of length 1, while (d1, d2) demonstrates MAR with 2 output intervals.

In this work, we present *Set AutoRegressive Modeling* (SAR), extending causal learning by generalizing sequence order and output intervals to arbitrary configurations. Specifically, compared with AR that splits the training process into sub-processes that output one single token in fixed raster order, SAR is able to input token sequence in any order (some examples are illustrated in Fig. 3 and Fig. 5), and splits it into any number of token sets, each as a sub-process that output multiple tokens. In order to represent the sequential relationship of token sets, we introduce generalized causal masks. As shown in Fig. 2, the classical causal mask (a1) is a lower triangular matrix; when the set contains more than one token (b1, c1, d1), the matrix becomes block-wise and is called a generalized causal mask. Within our framework, we show that AR, VAR (analogously), and MAR emerge as special cases of SAR, with AR and MAR representing two extreme instances. Refer to the left side of Fig.2 and Table1 for conceptual illustrations. Moreover, by the new formulation, we offer a path for smoothly transiting between AR and MAR. The intermediate states of SAR enables one to train a few-step causal model in support of KV cache acceleration that inherits both the advantages of AR and MAR models. Given that classical AR models, such as the decoder-only transformer, fails in the SAR setting, we propose a simple model architecture termed *Fully Masked Transformer (FMT)*. FMT adopts the encoder-decoder structure proposed in the original transformer (Vaswani, 2017) to enable both recording the output position and facilitating position-aware interaction between seen and output tokens. It incorporates generalized causal masks into each attention process to keep the causal manner, and the details can be referred to Fig. 4.

Under the SAR framework with FMT, we conduct experiments to explore the properties of SAR on the ImageNet $256 \times 256$ benchmark. We examine the relationship between the two hyper-parameters—sequence order and output intervals—and their impact on model performance, few-step generalization ability, and inference order generalization ability, discussing the associated trade-offs. Then, we train a text-to-image model on 20 million high-aesthetic images to further validate the generation capability of the transition states in SAR. Using limited computational resources and data, our model demonstrates the capability to generate photo-realistic images of arbitrary aspect ratios that adhere to the text descriptions.

Our main contributions are:

    i) We propose Set AutoRegressive Modeling, that unifies existing AR variants and offers new states between the two extremes, AR and MAR. The new states enables the training of few-step causal generation models.

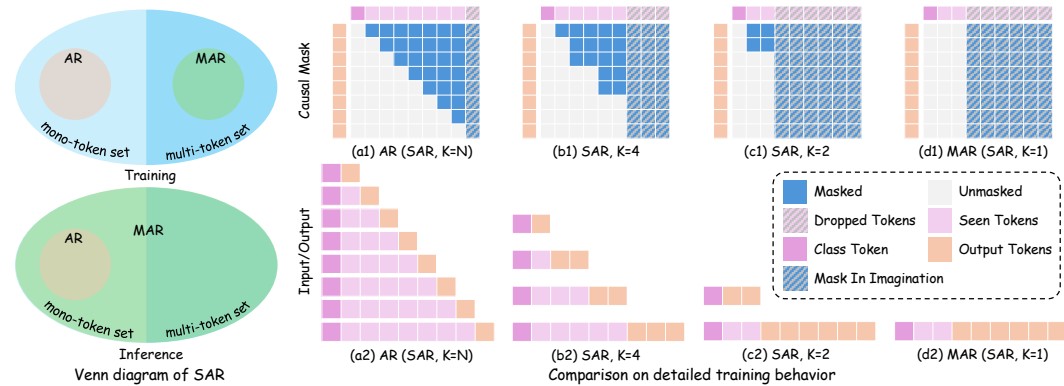

Figure 2: Conceptual illustration. SAR integrates existing AR variants by manipulating the sequence order and output intervals, creating a smooth transition path from classic AR to MAR.

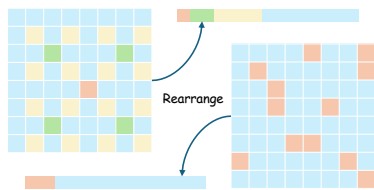

Figure 3: Sequence in any order can be rearranged as a causal one.

Table 1: Comparison among existing autoregressive image generation paradigms. SAR is more flexible and enjoys merits of other paradigms.

| Method | AR | VAR | MAR | SAR |
|---|---|---|---|---|
| Few-step inference | ✗ | ✓ | ✓ | ✓ |
| KV cache | ✓ | ✓ | ✗ | ✓ |
| Training/inference | Match | Match | Unmatch | Flexible |
| Common VAE | ✓ | ✗ | ✓ | ✓ |

ii) In line with SAR, we design a transformer model named Fully Masked Transformer, which enables causal learning with any sequence order and any output intervals.

iii) We conduct extensive experiments to investigate the properties of SAR and the modeling capability of FMT. With a particular focus on the transition states, we explore the effectiveness of text-to-image generation.

## 2 RELATED WORK

### 2.1 AUTOREGRESSIVE AND MASKED MODELING

Originated in language processing, GPT series (Radford, 2018; Radford et al., 2019; Brown, 2020) and BERT (Devlin, 2018) are representative works in autoregressive and masked modeling respectively. During the AR training, the current output token can only be observed by the preceding tokens. At inference tokens remain unchanged once output, facilitating the use of KV cache acceleration. Recently some work (Cai et al., 2024; Gloeckle et al., 2024) studies to reduce the inference steps by training multiple prediction heads and conducting speculative decoding (Leviathan et al., 2023; Chen et al., 2023) at inference. In contrast, BERT (Devlin, 2018) employs a bidirectional modeling approach known as masked modeling, to capture contextual information. It randomly masks a portion of tokens at a high masking ratio and trains the model to predict these masked tokens. At inference, BERT models can iteratively generate the output sequence with fewer steps than AR methods, at the cost of global calculation. Additionally, some works have introduced context perception into AR models. For example, XLNet (Yang, 2019) integrates insights from BERT by permuting the input sequence to enable bidirectional training with AR models. On image modality, our work not only provides further unification of AR and BERT models but also builds a smooth path connecting AR and BERT, where one can train models with both their merits.

### 2.2 AUTOREGRESSIVE IMAGE GENERATION

By tokenizing continuous images into discrete tokens using VQ-VAE (Van Den Oord et al., 2017; Razavi et al., 2019; Esser et al., 2021), image synthesis can be accomplished by AR models (Esser

Table 2: Some examples on SAR setting. $\mathrm{rand}(N, K)$ means randomly generate $K$ natural numbers, whose total sum is $N$.

| SAR | order | #sets | intervals |
|---|---|---|---|
| AR | raster | $N$ | $1, 1, 1, ...$ |
| VAR | custom | $log_4 N + 1$ | $1, 4, 16...$ |
| MAR | random | 1 | $\mathrm{rand}(N, 2)$ |
| Transition | random | K | $\mathrm{rand}(N, K)$ |

Table 3: Model setting of Fully Masked Transformer. The numbers of encoder and decoder layers are set equal for simplicity. Other configurations follows LlamaGen (Sun et al., 2024).

| SAR | Parameters | Enc. Layers | Dec. Layers | Width | Heads |
|---|---|---|---|---|---|
| B | 125M | 6 | 6 | 768 | 12 |
| L | 394M | 12 | 12 | 1024 | 16 |
| XL | 893M | 18 | 18 | 1280 | 20 |

et al., 2021; Lee et al., 2022; Ramesh et al., 2021; Yu et al., 2021; 2022; Liu et al., 2024; Luo et al., 2024) just like language modeling. Recently, Llamagen (Sun et al., 2024) verifies the generation capability of plain LLM, Llama (Touvron et al., 2023) on image modality. VAR (Tian et al., 2024) divides the image latent space into several scale groups by training a multi-scale VAE, and conduct next-scale prediction. Li et al. (2024) points out that the BERT-like image generation models (*e.g.*, MaskGIT (Chang et al., 2022), MagViT (Yu et al., 2023a;b), MAGE (Li et al., 2023), MAR (Li et al., 2024)) can also be regarded as autoregressive ones at inference, and as a result, we call BERT-like image generation models as MAR models. AutoNAT (Ni et al., 2024) revisits and improves the designs of training and inference of MAR models. Li et al. (2024) additionally shows that autoregressive image generation can also be conducted on continuous latent space with diffusion loss. Our proposed SAR paradigm can encompass the existing approaches as special instances, and provide the users with more flexible design space regarding various trade-offs. The supporting model of SAR is built upon LlamaGen (Sun et al., 2024) for its plain nature.

## 3 METHOD

In this section, we first review the AR and MAR paradigms. Then, we point out that conceptually these two methods differ in sequence order and output intervals, based on which we introduce Set AutoRegressive Modeling (SAR), and present the model design.

### 3.1 PRELIMINARY

**AutoRegressive Modeling (AR).** AR models the distribution of a token sequence $\{x^1, x^2, ..., x^n\}$ by the 'next-token prediction' objective defined as

$$p(x^1, ..., x^n) = \prod_{i=1}^{n} p(x^i | x^1, ..., x^{i-1}),\qquad(1)$$

where $p(x^1, x^2, ..., x^n)$ is the probability density function. Regarding the implementation, AR models are typically a decoder-only transformer with causal masks, as shown in Fig. 2 (a1). During training, the input to the model is set as the sequence shifted by one position, *i.e.*, dropping the last token, and padding a class token at the beginning (under the class-conditioned setting). The target is the original sequence, such that each output token is aligned with its 'next token'. At inference, the model can output tokens one by one in an autoregressive manner.

**Masked AutoRegressive Modeling (MAR).** MAR has recently been abstracted by Li et al. (2024), which describes the inference process of BERT-like (Devlin, 2018) image generation methods (Chang et al. (2022); Li et al. (2023); Yu et al. (2023a;b); Li et al. (2024)). In training, the input tokens are partially random masked with a high ratio (*e.g.*, $70\% - 100\%$ in Li et al. (2024)), and the model is trained to learn to predict the masked part. Fig. 2 (a2) and (d2) illustrate that AR trains $n$ sub-processes in a single iteration, while MAR processes one sub-process at a time. At inference, these methods can predict multiple tokens at once, costing less number of steps than AR models. However, because the masked modeling process is not causal, it cannot support causal techniques, *e.g.*, KV cache acceleration. Li et al. (2024) define 'next set-of-tokens prediction' as

$$p(x^1, ..., x^n) = p(X^1, ..., X^K) = \prod_{k=1}^{K} p(X^k | X^1, ..., X^{k-1}),\qquad(2)$$

**Algorithm 1** SAR Training

**Input:** Dataset $D$, Model $M$, Loss Function $\mathcal{L}$, Sequence Order $\texttt{od}$, Output Intervals $\texttt{intv}$
**Output:** Model $M$
**for** image code $x$, label $y$ **in** $D$ **do**
    $x \leftarrow \text{rearrange}(x, \texttt{od})$, $t \leftarrow x$
    $x \leftarrow \text{drop\_last}(x, \texttt{intv}[-1])$
    $x \leftarrow \text{concat}(y, x)$
    $m_e, m_{ds}, m_{dc} \leftarrow \text{gen\_masks}(\texttt{intv})$
    $o \leftarrow M(x, m_e, m_{ds}, m_{dc}, \texttt{od})$
    $l \leftarrow \mathcal{L}(o, t)$, backpropagate $l$
**end for**
**return** $M$

**Algorithm 2** SAR Inference

**Input:** Model $M$, Label $y$, Sequence Order $\texttt{od}$, Output Intervals $\texttt{intv}$
**Output:** Image Code $x$
$x \leftarrow \text{zero\_initialize}(\text{sum}(\texttt{intv}))$
$m_e, m_{ds}, m_{dc} \leftarrow \text{gen\_masks}(\texttt{intv})$
**for** $i$ **in** $\texttt{intv}$ **do**
    $o \leftarrow M(y, m_e, m_{ds}, m_{dc}, \texttt{od}, i)$
    $z \leftarrow \text{sample}(o)$
    $y \leftarrow \text{concat}(y, z)$
    $x \leftarrow \text{scatter}(x, z, \texttt{od}, i)$
**end for**
**return** $x$

where $X^k = \{x^i, x^{i+1}, ..., x^j\}$ is a *set of tokens* to be predicted at the $k$-th step. Eq. equation 2 generalizes vanilla next-token prediction Eq. equation 1 at inference time.

## 3.2 SET AUTOREGRESSIVE MODELING

**Sequence order and output intervals characterize autoregressive paradigms.** Actually, the token sequence in any output order can be rearranged into a causal one. AR is the simplest case whose input sequence is inherently causal. The other two instances with respect to an $8 \times 8$ image token grid are shown in Fig. 3. The left order is derived by downsampling the tokens using nearest neighbor interpolation (so the token value stays unchanged after interpolation). We make the model progressively output tokens downsampled with a scale factor of $1/8$, $1/4$, and $1/2$, and finally the rest of the tokens in a scale-aware order. It shares a similar spirit with VAR (Tian et al., 2024), so we call it a 'next-scale' variant. In this case, we can rearrange the tokens in the scale order. The right subfigure corresponds to mask modeling. By putting the unmasked tokens at the front and masked ones as the rest, we also derive a causal sequence.

Next, we consider the output intervals. For example, the output intervals of the 'next-scale' variant in Fig. 3 are $1, 4, 16, 43$, while those of the masked variant are the number of masked tokens and unmasked tokens. Since these variants output multiple tokens in each interval, they should be paired with generalized causal masks in training. Some conceptual instances are shown in Fig. 2 (b1, c1, d1), where generalized causal masks extend the classical causal mask (a1) to a block-wise format. The generalized causal mask can be uniquely determined by the output intervals.

SAR generalizes AR by extending the sequence order and the output intervals to any possible scenarios. In Fig. 2 (a1, d1) we can see that the causal mask of AR and MAR are two extreme case. In the intermediate states of SAR, one can train causal models with few-step inference enabled, which do not appear in either AR or MAR families. For example, if a 8-token sequence is split into 4 sets with $1, 2, 2, 3$ tokens, the causal mask should be like that in Fig. 2 (b1). In short, SAR extends 'next-set prediction' in Eq. 2 to the training phase.

**The model implementation—Fully Masked Transformer.** The realization of SAR is not straightforward, though. Classical AR models, *e.g.*, the decoder-only transformer fails in three aspects. i) When AR shifts the sequence to align the current set with the previous set, it will find the number of tokens may not be equal. ii) AR models can only model the output-seen relations with fixed and simple 'next token' forms of relative positional relationships, rendering them ineffective in complex scenarios involving arbitrary sets. iii) Given a token at a specific position, AR models output it based on its relative steps to the first token, leading to failure when outputting arbitrary sets. These drawbacks inspire the design philosophy: i) the model should have perception of absolute positions for outputting arbitrary token sets, and ii) the output tokens and the seen tokens should be placed into two containers, each with positional encoding, to facilitate their position-aware interaction.

Hence, we split the decoder-only transformer into two parts, an encoder and an decoder. The encoder takes in the image tokens and extract the semantic features. The decoder records the output position with position embeddings and models the interaction between output tokens and seen tokens from the encoder, at the cost of adding cross-attention in each decoder layer.

Additionally, generalized causal masks are added into each attention process, in the spirit of 'the current token set to be predicted can only see preceding sets'. In short, it can be regarded as a vanilla encoder-decoder transformer (Vaswani, 2017) with generalized causal masks in all attention processes. Consequently, we refer to it as the Fully Masked Transformer (FMT). Due to the fully causal feature, FMT naturally supports causal techniques like KV cache acceleration.

**The training procedure.** In order to train one model under the SAR framework, one should first specify the hyper-parameters, sequence order and output intervals. Based on the order setting, we first rearrange the sequence to the causal version (Fig. 3). And we set the target as the rearranged causal sequence. Next, based on the output intervals, we drop the last set of the rearranged sequence and prepend a class token. The resulting sequence is then fed into the encoder. Then the model can be trained with the common cross entropy loss. We list several combinations of sequence order and output intervals in Table 2, where we also add the number of sets for better understanding. The overall training procedure is illustrated in Algorithm 1.

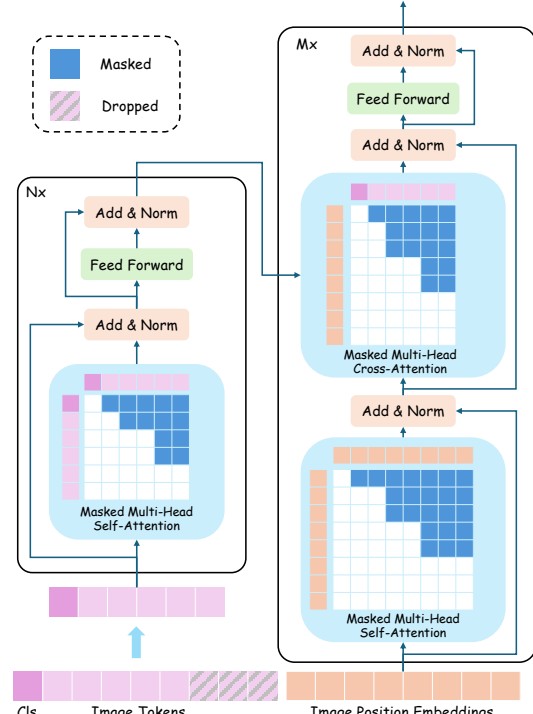

Figure 4: The model architecture of Fully Masked Transformer. Conceptually, it is the transformer in Vaswani (2017) plus generalized causal masks.

**The inference configuration.** Since our work is a generalized AR framework, SAR naturally supports advanced strategies developed for AR models, such as top-k, top-o, and min-p sampling. In this work, we directly apply some simple strategies for inference; one may also customize their own inference schedules. The inference algorithm is summarized in Algorithm 2.

## 4 EXPERIMENTS

### 4.1 EXPERIMENTAL SETTINGS

We conduct exploratory experiments on ImageNet (Deng et al., 2009) $256 \times 256$ benchmark. We use the tokenizer provided by Sun et al. (2024), and precompute the image codes before training as in Sun et al. (2024). We always use a batch size of $256$ and learning rate of $1e-4$ during training. Models in the transition states SAR-TS in Table 4 is trained for 300 epochs, while all other models are trained for 200 epochs. Other training settings follow Sun et al. (2024). For evaluation, we report the common used FID (Heusel et al., 2017), IS (Salimans et al., 2016), Precision and Recall metrics. Unless otherwise specified, the default setting is cfg=2.0, top-k=0 (all), top-p=1.0, temperature=1.0. The evaluation is conducted following Dhariwal & Nichol (2021).

### 4.2 HYPER-PARAMETERS IN SAR

**Configuration on sequence order and output intervals for training.** We test several hyper-parameter combinations containing some common settings and two customized ones named 'next-scale' and 'masked modeling'. Among the common settings, we control the sequence order, the output schedule, and the number of sets, where the latter two jointly determine the output intervals. There are six choices in order.

The first four is shown in Fig. 5. (a) The 'raster' order is the classical AR order, while (b) is its reversed version. (c) and (d) are the 'Swiss roll', clockwise, from outside to inside and from inside to outside respectively. The other two are fixed-random and random. The former means that we randomly generate an order and fix it during training, while the latter indicates online random.

There are two types of output schedules involved, which can determine the output intervals based on the number of sets as follows: i) cosine: given a set number $K$, the output intervals $\{n_i\}_1^K$ follows $n_i = [N(\cos(\frac{\pi}{2}\frac{i}{K}) - \cos(\frac{\pi}{2}\frac{i-1}{K}))]$, as in Li et al. (2024). Note: here at least one token is ensured to be output at each

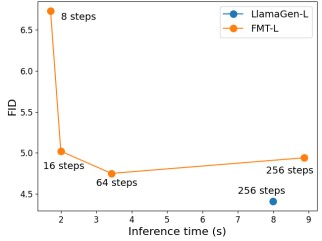

Figure 5: Some sequence order settings in the experiment. Taking the $8 \times 8$ case as illustration.

step, thus given sequence order as raster and step number as $256$, it will recover to AR. ii) random: given a set number $K$, we randomly generate $K-1$ natural numbers (there may be equal numbers) between $0$ and $N$ with the same probability, such that the sequence can be split into $K$ intervals by these partition numbers. Under the common settings, we conduct experiments in the format of (sequence order)-(number of sets)-(output schedule). For example, raster-64-cosine indicates a raster-order sequence with $64$ sets under a cosine schedule.

The customized settings includes i) next-scale: we rearrange the $16 \times 16$ image tokens such that the 1st set contains the $1/16$ nearest neighbor downsampled token, the 2nd set contains the four $1/8$ downsampled tokens, ..., the 5th set contains the rest of tokens, as illustrated on the left of Fig. 3, and ii) masked modeling: we follow the settings in Li et al. (2024). Actually it can be derived by removing the loss of the first token set and modifying the random strategy in 'random-2-random'.

**Configuration on model size of FMT.** The implementation of FMT is based on the GPT model in LlamaGen (Sun et al., 2024). For simplicity, we do not adopt the asymmetric design in He et al. (2022), but just divide the $N$-layer transformer into an encoder and a decoder, each with an equal number of layers. One can refer to Table 3 for detailed model configurations. Compared with LlamaGen, we add an extra cross-attention module at each decoder layer, so under the same model size, the number of parameters of FMT is slightly larger.

### 4.3 MAIN RESULTS

Table 4 presents a comprehensive comparison of performance across various methods and models, where we train models for each AR setting within the SAR paradigm.

**SAR as AR.** The raster-256-cosine variant of SAR recovers to conventional AR. We evaluate the performance of FMT-B, FMT-L, and FMT-XL trained for 200 epochs, with the results presented in Table 4. Under the same setting (stared in Table 4), FMT outperforms LlamaGen under the same model size.

**SAR as MAR.** SAR recovers to MAR under the 'masked modeling' setting. The performance of FMT is also shown in Table 4.

**SAR as VAR, analogously.** By customizing the sequence order and output intervals as 'next-scale', illustrated on the left side of Fig. 3,

Figure 7: Trade-off between performance and time, using LlamaGen-L as a reference.

we derived a rough variant of VAR. The results are presented in Table 4. While this serves primarily as a conceptual example, its performance lags significantly behind that of VAR (Tian et al., 2024).

**Transition states of SAR.** The last three rows of Table 4 present the performance ($64$ steps) of a specific design choice in the transition states of SAR, which will be detailed in the ablation study. Compared to FMT under the AR configuration, the performance in this case is somewhat lower. However, models trained under this setting can generalize across inference steps and orders while maintaining their causal features. A straightforward merit is that, we can enable KV cache acceleration while performing few-step inference. A diagram on performance-time trade-off is shown in Fig. 7, where the inference time is tested by generating a batch of $8$ images on one A100 GPU. And of course, we can also apply other causal techniques to promote the performance or efficiency.

### 4.4 ABLATION STUDY

**Varying sequence orders in training/inference.** Table 5 presents the results obtained by fixing the output intervals to $1, 1, \ldots$ while training and inferring with various sequence orders. It is clear that

Table 4: Performance comparison among various paradigms and models. '-re' means rejection sampling. For LlamaGen (Sun et al., 2024), * means direct training on $256 \times 256$ images; otherwise, training is on $384 \times 384$ and the output is resized in evaluation. 'TS' denotes transition state.

| Type | Model | #Params | FID↓ | IS↑ | Precision↑ | Recall↑ |
|---|---|---|---|---|---|---|
| | BigGAN (Brock, 2018) | 112M | 6.95 | 224.5 | 0.89 | 0.38 |
| GAN | GigaGAN (Kang et al., 2023) | 569M | 3.45 | 225.5 | 0.84 | 0.61 |
| | StyleGAN-XL (Sauer et al., 2022) | 166M | 2.30 | 265.1 | 0.78 | 0.53 |
| | ADM (Dhariwal & Nichol, 2021) | 554M | 10.94 | 101.0 | 0.69 | 0.63 |
| Diffusion | CDM (Ho et al., 2022) | - | 4.88 | 158.7 | - | - |
| | LDM-4 (Rombach et al., 2022) | 400M | 3.60 | 247.7 | - | - |
| | DiT-XL/2 (Peebles & Xie, 2023) | 675M | 2.27 | 278.2 | 0.83 | 0.57 |
| | MaskGIT (Chang et al., 2022) | 227M | 6.18 | 182.1 | 0.80 | 0.51 |
| | MaskGIT-re (Chang et al., 2022) | 227M | 4.02 | 355.6 | - | - |
| Masked AR | MAGE (Li et al., 2023) | 230M | 6.93 | 195.8 | - | - |
| | MAR-H (Li et al., 2024) | 943M | 1.55 | 303.7 | 0.81 | 0.62 |
| (SAR, K=1) | FMT-B | 125M | 6.98 | 222.28 | 0.87 | 0.36 |
| | FMT-L | 394M | 6.13 | 278.81 | 0.88 | 0.40 |
| VAR | VAR-d30-re (Tian et al., 2024) | 2.0B | 1.80 | 356.4 | 0.83 | 0.57 |
| (SAR, customized) | FMT-B | 125M | 12.49 | 148.53 | 0.76 | 0.36 |
| | VQGAN-re (Esser et al., 2021) | 1.4B | 5.20 | 280.3 | - | - |
| | ViT-VQGAN-re (Yu et al., 2021) | 1.7B | 3.04 | 227.4 | - | - |
| | RQTran.-re (Lee et al., 2022) | 3.8B | 3.80 | 323.7 | - | - |
| | LlamaGen-B* (cfg=2.00) | 111M | 5.46 | 193.61 | 0.84 | 0.46 |
| | LlamaGen-L (cfg=2.00) | 343M | 3.07 | 256.06 | 0.83 | 0.52 |
| AR | LlamaGen-XL (cfg=1.75) | 775M | 2.62 | 244.08 | 0.80 | 0.57 |
| | LlamaGen-L* (cfg=2.00) | 343M | 4.41 | 288.17 | 0.86 | 0.48 |
| | LlamaGen-XL* (cfg=1.75) | 775M | 3.39 | 227.08 | 0.81 | 0.54 |
| (SAR, K=N) | FMT-B (cfg=2.00) | 125M | 5.40 | 216.93 | 0.87 | 0.42 |
| | FMT-L (cfg=2.00) | 394M | 3.72 | 297.54 | 0.86 | 0.49 |
| | FMT-XL (cfg=1.75) | 893M | 2.76 | 273.76 | 0.84 | 0.55 |
| | FMT-B (cfg=2.00) | 125M | 7.04 | 182.01 | 0.84 | 0.40 |
| SAR-TS | FMT-L (cfg=2.00) | 394M | 4.75 | 261.27 | 0.84 | 0.46 |
| (random-16-random) | FMT-XL (cfg=1.90) | 893M | 4.24 | 249.23 | 0.82 | 0.51 |

Table 5: FID results of training/inference with different order settings. The model is FMT-B.

| Training/inference | raster | reversed-raster | roll | reversed-roll | fixed-random | random |
|---|---|---|---|---|---|---|
| raster | **5.40** | 136.54 | 114.41 | 99.13 | 132.61 | 120.82 |
| reversed-raster | 133.18 | **6.01** | 123.47 | 118.67 | 146.48 | 138.29 |
| roll | 81.93 | 114.23 | **6.93** | 133.50 | 130.28 | 117.69 |
| reversed-roll | 125.78 | 134.25 | 155.04 | **6.44** | 128.62 | 125.56 |
| fixed-random | 104.24 | 117.23 | 116.58 | 103.03 | **7.49** | 86.90 |
| random | 22.95 | 22.91 | 13.66 | 10.32 | 7.83 | **7.76** |

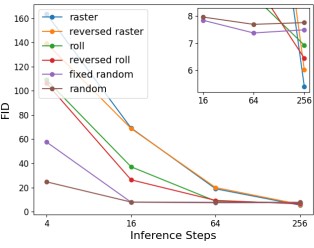

Figure 6: Effect of order.

although position embeddings are used, a fixed sequence order typically does not allow the model to generalize across different inference orders.

**Fixed few-step generation.** By fixing the sequence order to the raster order and using a cosine schedule for the intervals, we investigate few-step SAR training by varying only the number of sets. As illustrated on the left of Fig. 8, we observe that, i) since both the order and the schedule are fixed, the best inference performance typically occurs when the number of sets used at inference matches that used in training; ii) from the inset in the upper right, it is evident that only the 64-set configuration is effective for few-step generation, while the others significantly degrade performance.

**Randomness in orders enables few-step generalization.** We fix the number of sets at 256 and the interval schedule to $1, 1, \ldots$, varying only the sequence order. As shown in Fig. 6, models trained with the raster, reversed raster, roll, and reversed roll orders struggle to generalize to few-step generation. In contrast, models trained with a random order demonstrate good generalization across inference steps, albeit at the cost of lower FID scores (5.40 FID with raster order vs. 7.76

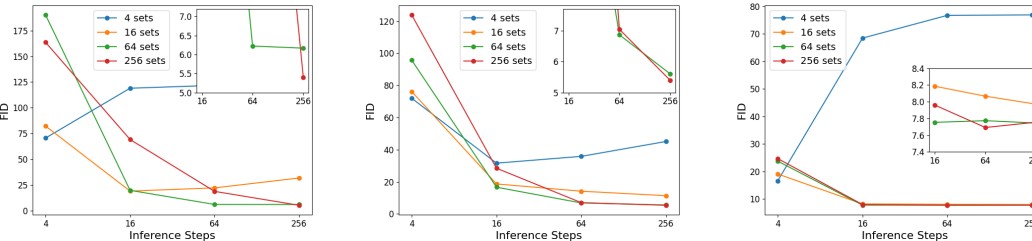

Figure 8: Effect of set numbers when training SAR with (left) raster order and cosine schedule, (middle) raster order and random schedule, and (right) random order and cosine schedule.

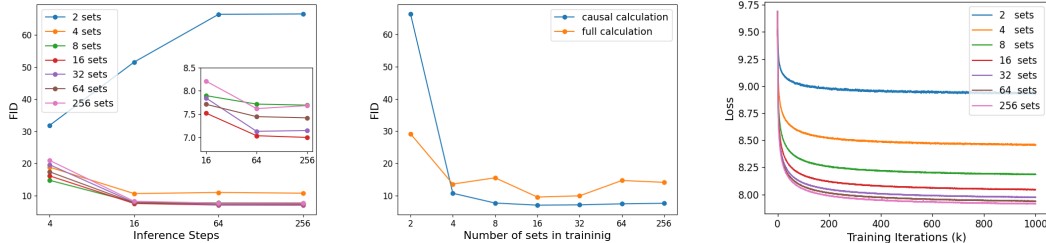

Figure 9: Exploration when sequence order and output schedule are both set as random. Left: Performance wrt. number of sets. Middle: After causal training, comparison between causal and full attention calculation. Right: Training loss of various set numbers.

FID with random order). It may be surprising that fixing a randomly generated order during training can achieve similar generalization ability to that of a fully random order.

**Random output intervals enables few-step generalization.** We fix the sequence order to raster and use a random schedule with varying numbers of sets. The results on the middle of Fig. 8 indicate that when the number of sets is large (*e.g.*, 64 or 256), random intervals facilitate few-step generalization.

**The relationship between number of sets and causal learning.** Under the setting of random sequence order, we examine performance in relation to the number of sets. Figures 8 (right) and 9 (left) show the results with cosine and random output schedules, respectively. We observe that, with a large number of sets, performance remains stable; however, it declines significantly when the set number decreases to 4 in the cosine case and 2 in the random case. Intuitively, to develop a causal model, the model must be trained to predict sets one by one, with more sets indicating a greater degree of causality. If the number of sets is too small, the model struggles to learn causal relationships effectively. Another interesting observation is that, after trained with small number of sets, abandoning causality can help restore performance. As shown in the middle of Fig. 9, the performance of the model trained with 2 sets gets better when replacing the causal attention with full attention. However, model trained with other set numbers cannot benefit from full attention, because they receive more sufficient causal learning. The last subfigure of Fig. 9 illustrates the loss curves during training, where the level of loss may be regarded as a measure of training difficulty. The loss of the best-performing configuration, 16 sets, is situated at a mid-level.

**Further discussion on the MAR setting of SAR.** There are some details that need to be clarified. i) In Sec. 4, we mentioned that the MAR setting is derived based on 'random-2-random' by only supervising the second set, and using the random strategy in Li et al. (2024). From Table 6, Row 1 vs. Row 2 tells us that, with the same model, removing the loss of the first set has little impact on model training; not removing it may even lead to better performance. This fact demonstrates that the transition from $K = 2$ to $K = 1$ (*i.e.*, MAR) in SAR is smooth. ii) It is worth noting that, in the MAR case the generalized causal masks in the encoder self-attention and decoder cross-attention is equivalent to having none. And only the causal mask in decoder self-attention will affect the training. Intuitively, there is no need to prepare causal mask in training because at inference MAR always conduct global attention. Row 1 vs. Row 3 in Table 6 indicates that the existence of causal mask in decoder self-attention hurts the performance. iii) Row 4 is a setting from Fig. 9. The large discrepancy in performance between Row 2 and Row 4 emphasizes the importance of proper

Table 6: Relationship between performance and detailed MAR settings. The inference process is BERT-like, with full attention.

| Row | Random Strategy | K | Causal Mask in Decoder Self-Attn | FID↓ | IS↑ | Precision↑ | Recall↑ |
|---|---|---|---|---|---|---|---|
| 1 | MAR (Li et al., 2024) | 1 | ✓ | 8.81 | 148.36 | 0.76 | 0.46 |
| 2 | MAR (Li et al., 2024) | 2 | ✓ | 7.19 | 183.31 | 0.83 | 0.39 |
| 3 | MAR (Li et al., 2024) | 1 | ✗ | 6.98 | 222.28 | 0.87 | 0.36 |
| 4 | Equal Probability | 2 | ✓ | 29.20 | 46.91 | 0.65 | 0.52 |

random strategy. This also suggests that our strategy for SAR transition states may not be optimal, which could explain the sub-optimal SAR-TS results in Table 4.

## 4.5 APPLICATION: TEXT-TO-IMAGE GENERATION

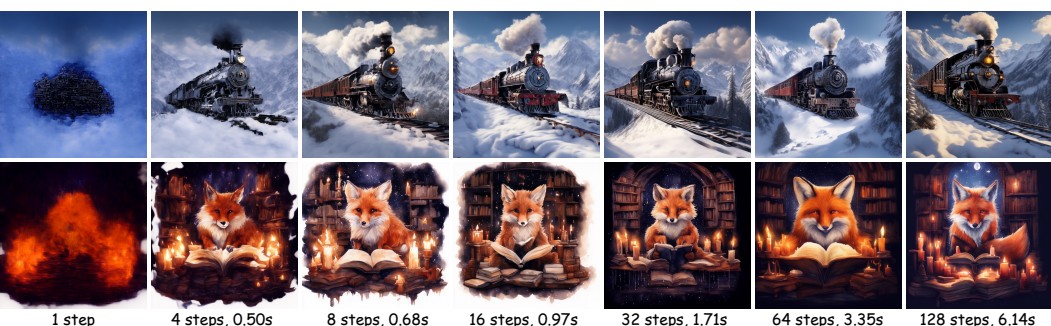

| 1 step | 4 steps, 0.50s | 8 steps, 0.68s | 16 steps, 0.97s | 32 steps, 1.71s | 64 steps, 3.35s | 128 steps, 6.14s |

Figure 10: Step number and time cost of Lumina-SAR at $1024 \times 1024$ (Full 4096 steps cost $187.8s$).

We leverage the FMT-XL model for text-to-image (T2I) generation. The sequence order and the output schedule are set as random, the best practice with random order in ImageNet experiments. We adopt the training strategy with multiple aspect ratios enabled in Gao et al. (2024); Zhuo et al. (2024) and the multi-stage policy in Zhuo et al. (2024); Sun et al. (2024); Chen et al. (2024). Specifically, we set the number of sets as 16 and the base resolution as $256 \times 256$ in the first stage, and gradually increase the number of sets and the base resolution by a factor of 2. The final resolution is $1024$. At each training stage, we group images with different aspect ratios but similar resolutions, which are further padded to the same length. As for the language part, we adopt the Gemma-2B (Team et al., 2024) as the text encoder and concatenate the text embedding with the image tokens, with the conventional causal mask like that in Fig. 2 (a1). Other training settings including text-image training data are following Zhuo et al. (2024), and we name our T2I model as Lumina-SAR. As visualized in Fig. 1, Lumina-SAR can flexibly produce photo-realistic images in arbitrary resolutions.

We examine the time cost of Lumina-SAR for generating one image using one A100 GPU, as illustrated in Fig. 10. We observe that Lumina-SAR begins to produce acceptable images at around 4 to 8 steps. With 64 to 128 steps, it can deliver high-quality outputs, requiring a processing time of only 3 to 6 seconds. Typically, the full 4096 steps take 56 times longer than that required for 64 steps.

## 5 CONCLUSION

In this work, we propose Set AutoRegressive Modeling (SAR), a new AR paradigm that enables users to freely customize the AR training and inference processes. For SAR, we also develop a preliminary model architecture called the Fully Masked Transformer. We carefully explore the properties of SAR, with a particular focus on the intermediate states, which facilitate training models capable of both few-step generation and KV cache acceleration. Additionally, we train a T2I model under the SAR paradigm to validate the generation capabilities at the transition states of SAR.

**Limitation.** As a newly emerging paradigm, the exploration of SAR in this paper is limited, particularly concerning the performance of SAR intermediate states on ImageNet. Future work may focus on developing better training and inference schedules, designing model architectures that are more compatible with SAR, and exploring the application of SAR across additional modalities.

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

# A  APPENDIX

## A.1  MORE VISUALIZATIONS ON IMAGENET.

For $256 \times 256$ image generation on ImageNet, we generate some random samples that are not cherry picked. Fig. 11 and Fig. 12 exhibit samples produced by FMT-XL trained under the random-16-random and raster-256-cosine settings, respectively.

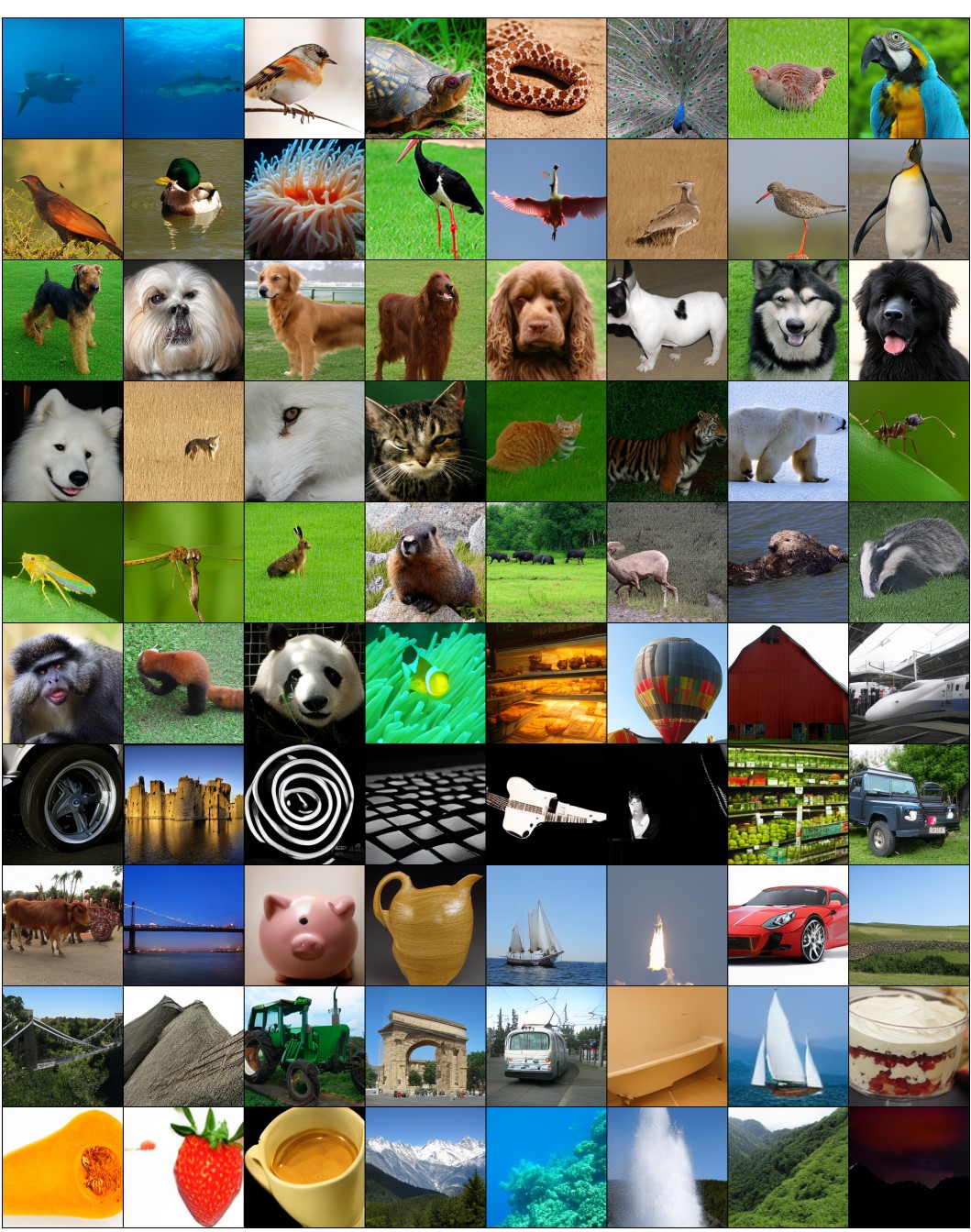

Figure 11: Samples generated by FMT-XL trained with SAR, random-16-random.

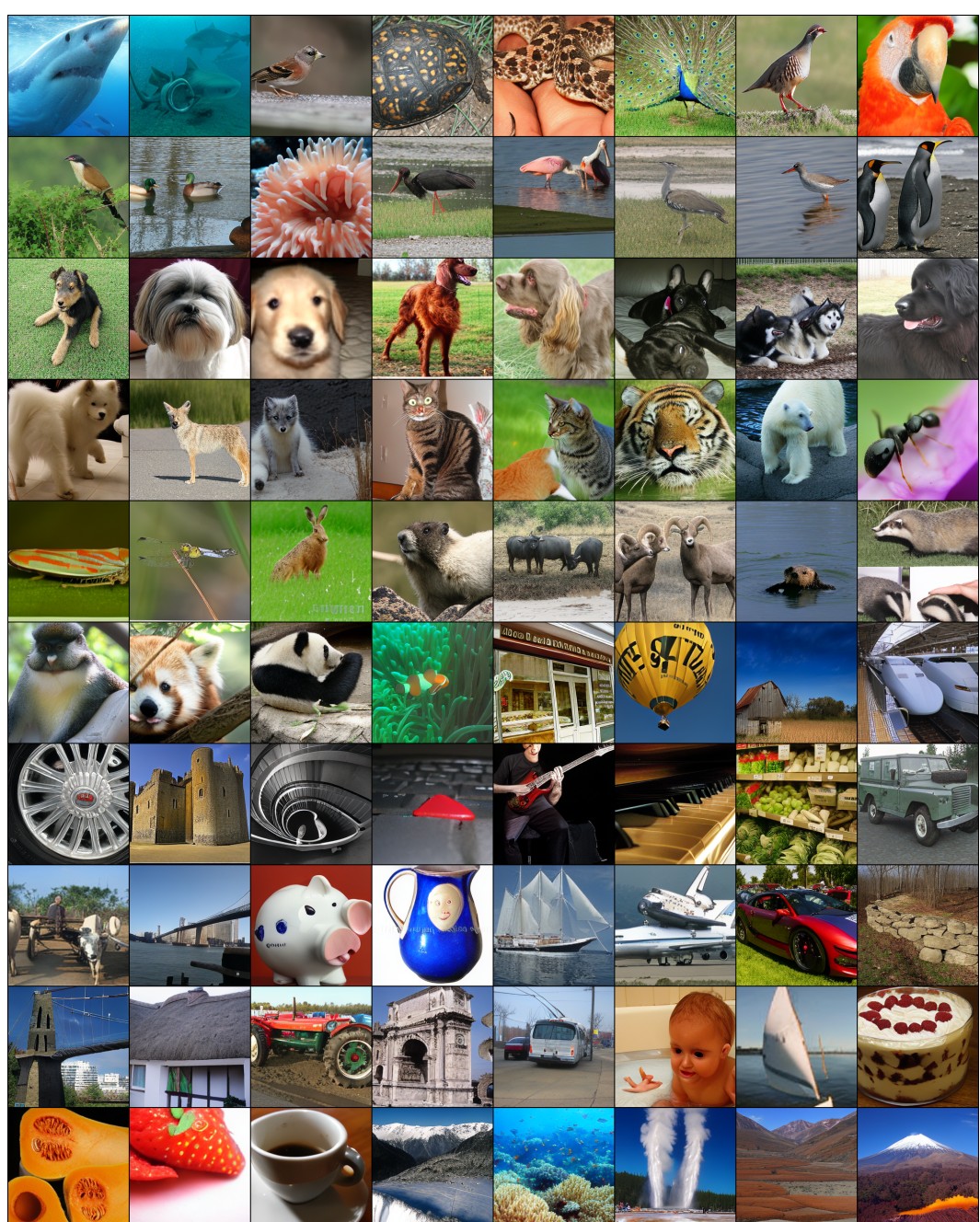

Figure 12: Samples generated by FMT-XL trained with SAR, raster-256-cosine (*i.e.*, classical AR).

## A.2 MORE VISUALIZATIONS ON T2I IMAGE SYNTHESIS

We provide additional visualizations generated by Lumina-SAR, and show them in Fig. 13.

## A.3 DETAILS ON FULLY MASKED TRANSFORMER

The position embedding as input to the decoder can be either learnable or fixed, such as sine embedding. In class-conditioned generation, we use learned embedding as in Li et al. (2024). In the T2I model, we use sine embedding to accommodate training with multiply aspect ratios: after each input image is fed into FMT, we first generate its sine embedding. Similar to LlamaGen (Sun et al., 2024),

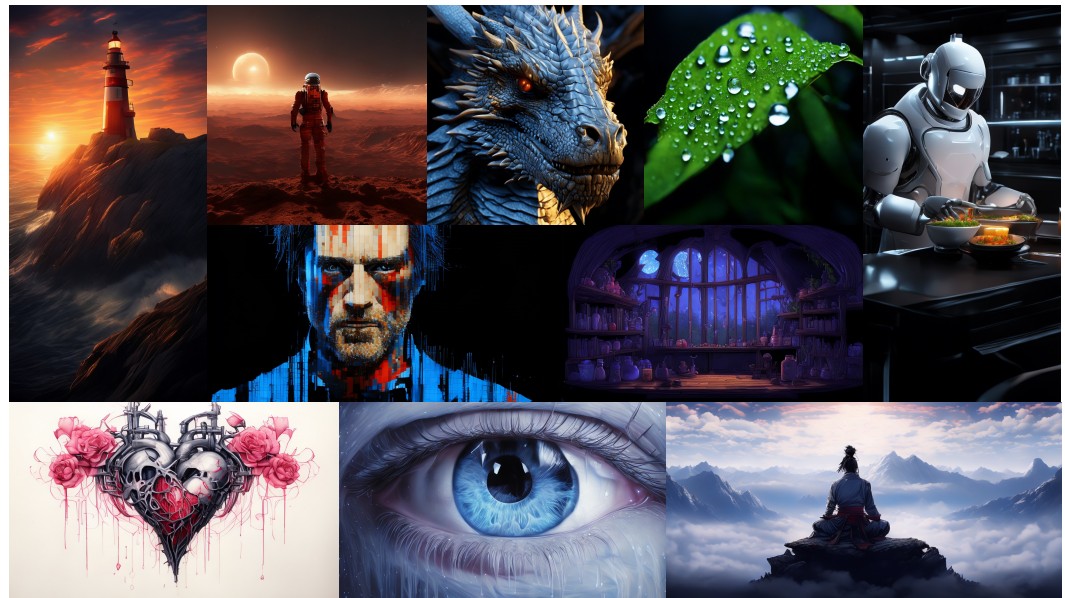

Figure 13: Samples generated by Lumina-SAR. The model is FMT-XL trained under the random-$x$-random setting of SAR, where $x$ is set as 16, 32 and 64 at the stage of $256 \times 256$, $512 \times 512$ and $1024 \times 1024$ respectively.

we use RoPE Su et al., 2024 to enable the position-aware interaction. Both the position embedding and RoPE are rearranged like what is done to the input tokens according to the sequence order, such that the positions are aligned.

## A.4 THE PERFORMANCE WRT. EVALUATION CONFIGURATIONS

We provide the results when adjusting the scale of classifier-free guidance and the top-k values in Fig. 14, where we use FMT-L trained under the random-16-random setting for 300 epochs and the number of sampling steps is set to 64. We observe behavior in SAR that is similar to that of classical AR models.

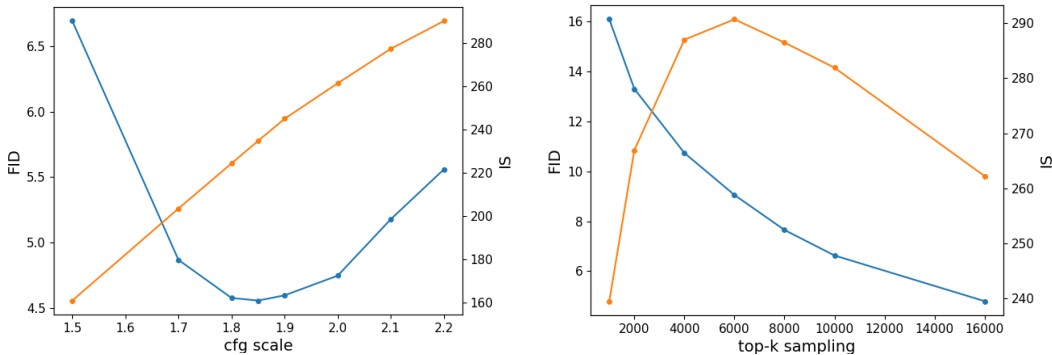

Figure 14: The effect of cfg scale (left), and top-k sampling (right).

## A.5 AN ISSUE ON THE GENERATION OF SAR-TS ON IMAGENET

We found that the SAR-TS models frequently encounter framing misalignment issues when generating images, which may be the reason for its higher FID scores. Some randomly generated examples are shown in Fig. 15. In a simultaneously generated batch of 8 images, the first, third, fifth, seventh and eighth exhibit this issue.

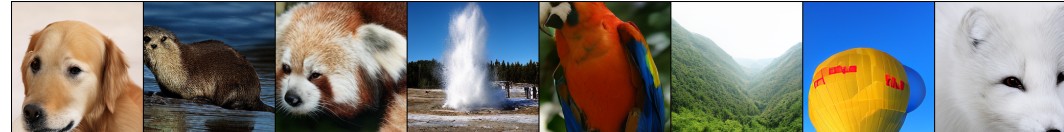

Figure 15: The framing misalignment issue on generated samples by SAR-TS models.

