# OpenReview forum: "Customize Your Visual Autoregressive Recipe with Set Autoregressive Modeling"
_ICLR.cc/2025/Conference — Submitted to ICLR 2025_

### Official Review · Reviewer_Zc8J · 2024-10-31

**Soundness:** 3
**Presentation:** 2
**Contribution:** 3
**Rating:** 3
**Confidence:** 3

**Summary:**

This paper introduces a new image generation method called SAR, which extends the traditional autoregressive model to a "next-set" framework. SAR enables flexible sequence ordering and output intervals, thus encompassing both the autoregressive and masked autoregressive models as specific cases. Additionally, the authors propose the Fully Masked Transformer architecture for SAR. Quantitative and qualitative results are presented on class conditional and text-to-image generation.

**Strengths:**

1. SAR is a flexible formulation and unifies the existing autoregressive and masked autoregressive models. A unified formulation is interesting.
2. The authors present an effective, new Fully Masked Transformer architecture for SAR.
3. The authors systematically studied the impact of factors such as sequence order and output interval on generation quality. I believe these experimental results provide valuable insights for advancing the field.

**Weaknesses:**

1. The authors claim that SAR unifies autoregressive and masked autoregressive models; however, SAR's FID lags significantly behind masked autoregressive models.
2. The authors note that SAR can utilize KV-cache acceleration, whereas masked autoregressive models cannot, and they claim this as an advantage of SAR. However, they do not provide any speed comparisons between SAR and masked autoregressive models. Instead, they only compare generation speed with LlamaGen, which is insufficient. Moreover, it raises questions as to why SAR’s FID is worse than LlamaGen’s in Figure 7, despite both models having a similar number of parameters.
3. The Text-to-Image generation experiment lacks proper evaluation.

**Questions:**

1. What does $N$ represent in Line 330 and Line 334? It seems that $N$ refers to the number of tokens in an image based on the notations in Figure 2 (a1), but the authors use $n$ to denote the number of tokens in an image in Equation (1).

2. In Line 342, the authors state that 'random-2-random' indicates the MAR setting. However, as shown in Figure 2 (d1), $K=1$ for MAR. I'm a bit confused; could you please provide further clarification?

3. What do $m_e$, $m_{ds}$ and $m_{dc}$ mean in Algorithm 1 and Algorithm 2?

4. When training with a fixed number $K$, how does SAR inference when the sampling steps differ from $K$?

5. I am concerned about the scaling properties of the encoder-decoder network structure. I recommend that the authors explore the scaling behavior of SAR or even scaling laws in their future work.

---

> ### Author Response · Authors · 2024-11-19
>
> # Reply to Reviewer Zc8J
>
> We thank the reviewer for valuable feedback. Below, we address the concerned points:
>
> ## 1. Possible Reason for Suboptimal FID of SAR-TS
>
> Actually, we find that the image quality of SAR-TS is good, but there are serious framing misalignment issues. For example, the randomly generated 8 images in Figure 15 in the Appendix. Among large number of generated images, the subjects frequently encounter problems of extremely large, extremely small, or only part displayed. We think it leads to the suboptimal FID.  Further, the issue may come from the strong reliance of SAR-TS on position embedding. Classical AR methods rely on static relative position relation, i.e., the next-token relation, so AR does not rely on position embedding. But SAR-TS uses random order, it perceives position completely from position embedding. It also explains that SAR-AR gets better results than the AR baseline, but falls behind when using random orders.
>
> ## 2. Training/Inference Speed Gain from KV Cache
>
> For a comprehensive comparison, we examine the time cost of FMT-XL (for text-to-image generation) when generating one 1024x1024 image using one A100 GPU. The generation order of AR is raster, while those of MAR and SAR are both random (actually the order does not affect speed). The results are as follows:
> | Setting | KV cache | 64 steps | 128 steps | 4096 steps |
> |---------|----------|-----------|------------|-------------|
> | AR | ✓ | - | - | 174.49s |
> | MAR | ✗ | 9.66s | 19.22s | 685.77s |
> | SAR-TS | ✗ | 7.45s | 14.72s | 606.35s |
> | SAR-TS | ✓ | 2.82s | 5.78s | 174.49s |
>
> We empirically find that there is no visual difference between using 64 steps and more. And, the visualization in Figure 1 and Figure 13 (in the Appendix) are all sampled with **64** steps. We can see that the inference speed of SAR is > 60 times than that of AR with KV cache enabled. Notably, in the transformer decoder, MAR applies global attention across all tokens, while the number of tokens processed in AR and SAR-TS increases gradually. Consequently, even without KV cache, the inference time for SAR-TS is shorter than that of MAR; when KV cache is enabled, SAR-TS is three times faster than MAR with $64$ or $128$ steps.
>
> ## 3. Performance of Text-to-Image Generation
>
> We test the performance with 64 steps and compare it with other existing methods/models. The results are as follows, where the test dataset is MJHQ-30K [1], results of other methods in the table is cited from SANA [2], and the FID metric and CLIP score is calculated using [3] and [4] respectively.
> | Model                            | Params (B)|  FID     | CLIP score |
> |----------------------------------|----------|----------|------------|
> | SDXL [5]       |   2.6    | 6.63     | 29.03      |
> | PlayGroundv2.5 [1]|   2.6    | 6.09     | 29.13      |
> | SD3-medium [6]  |   2.0    | 11.92    | 27.83      |
> | FLUX-dev [7]            |   12.0   | 10.15    | 27.47      |
> | Ours                             |   0.9    | 14.25    | 31.23      |
> From the table we can see that the FID of ours is worse than other methods, which is aligned with the results on ImageNet-256, while its CLIP score is better. For limitted resources, our T2I model is only trained for 200k iterations with batchsize of 256, which can be one of the reasons for worse FID.
>
> [1] Li et al. Playground v2.5: Three Insights towards Enhancing Aesthetic Quality in Text-to-Image Generation
>
> [2] Xie et al. SANA: Efficient High-Resolution Image Synthesis with Linear Diffusion Transformers
>
> [3] https://github.com/mseitzer/pytorch-fid
>
> [4] https://github.com/Taited/clip-score
>
> [5] Podell et al. SDXL: Improving latent diffusion models for high-resolution image synthesis
>
> [6] Esser et al. Scaling rectified flow transformers for
> high-resolution image synthesis
>
> [7] Black Forest Labs. Flux, 2024. URL https://blackforestlabs.ai/
>
> ## 4. Questions
> i) Thanks for indicating this. The $n$ in Eq. (1) is a typo.
> ii) 'random-2-random' means random order, 2 sets, random intervals. During training, MAR randomly divides tokens into 2 sets with random order, but it only calculates loss on the masked part. Therefore we said that ‘it can be derived by
> removing the loss of the first token set in ‘random-2-random’.
> iii) m_e, m_{ds}, m_{dc} mean causal masks for encoder self-attention, decoder self-attention and decoder cross-attention.
> iv) There is no strong relation in training and inference in SAR. One can adjust the output intervals to achieve sampling with any number of steps as in Algorithm 2.
> v) Thanks for your suggestion and we will explore the scaling behavior in the future.

---

> > ### Comment · Reviewer_Zc8J · 2024-11-21
> >
> > Thank you for your response! The response does not address my questions.
> >
> > 1. I believe the "framing misalignment" is due to cropping used in data processing. Other generative models also adopt image cropping techniques, and I don’t think this is a reliable explanation for high FID.
> >
> > 2. The authors claim that "the image quality of SAR-TS is good"; however, they do not provide any evidence. This is highly unprofessional.
> >
> > 3. The table in response 2 is also highly unprofessional. A rigorous approach would be to report the relationship between sampling time and FID. From your table, we cannot conclude that SAR produces images of the same quality faster.
> >
> > 4. FID is an outdated metric for text-to-image generation. More comprehensive metrics, such as Geneval [1] or T2i-CompBench [2], should be used instead. This is not my primary concern, as I understand that time during the public discussion phase is limited.
> >
> > 5. I understand that training text-to-image models requires a significant amount of computation. Therefore, under limited computational resources, I suggest comparing AR or MAR within the same setting, which is more appropriate than comparing them with SOTA models.
> >
> > I find the authors' response not only inadequate in addressing my concerns but also lacking in rigor. As a result, I have adjusted my score to 3.
> >
> > [1] Ghosh et al. Geneval: An object-focused framework for evaluating text-to-image alignment. NeurIPS 2023
> >
> > [2] Huang et al. T2i-compbench: A comprehensive benchmark for open-world compositional text-to-image generation. ICCV 2023

---

### Official Review · Reviewer_rmFX · 2024-11-02

**Soundness:** 3
**Presentation:** 3
**Contribution:** 2
**Rating:** 5
**Confidence:** 4

**Summary:**

This paper proposes SAR as a unified framework of AR and MAR. The key components of SAR include the number of sets, the number of tokens per set and the sequence order. To enable flexible design for these components, this work further propose FMT, an encoder-decoder transformer that utilizes an encoder to output the embedding of prefix tokens and generate new tokens via decoder, guided by the positional embeddings. Experimental results are provided to compare SAR with existing methods and show how each component influence the performance of SAR.

**Strengths:**

1. SAR incorporate several important design choice in AR models, including the number of sets, the number of tokens per set and the sequence order. Additionally, SAR employs Fully Masked Transformer to address the challenges of using arbitrary choices.
2. The paper includes a substantial amount of experimental results:
    - SAR achieves better performance than LlamaGen and reproduces comparable performance with MAR by using appropriate design choices.
    - Extensive ablation studies illustrate the impact of each design choice on the performance-step trade-off.
    - Experiments on text-to-image generation are also conducted.

**Weaknesses:**

1. Further clarification is needed regarding the fundamental differences between SAR and existing methods. While SAR is presented as a flexible framework, its key components exhibit similarities to those of existing methods.
    - For the number of sets and set length: Fig.2 shows that SAR represents a smooth transition from AR to MAR, as it enables different numbers of sets and set lengths. However, as stated in the paper and based on my understanding, MAR [1,2] also supports different set lengths during inference to achieve a trade-off between steps and performance. The extreme case for MAR mentioned in the paper (i.e., K=1) occurs only during training. Moreover, VAR [3] also uses the set-prediction approach and generates tokens in a causal manner. Therefore, whether viewd from the perspective of extending the inference of MAR to training and using causal architecture, or from the perspective of extending VAR accommodate different numbers of sets and varying set lengths, the modifications in this paper seem to be straightforward. The authors need to clarify the essential differences between the "continuous new state that SAR offers between AR and MAR" and the aforementioned methods. Is the difference only lies in the introduction of a new architecture FMT?
    - For the sequence order: The most general form of sequence order, random order, has already been used by previous MAR works [2]. This paper seems to simply adopt this method.

2. Regarding experimental results:
    - My main concern is that the performance improvement of SAR shown in Tab.4 is relatively limited.
        - For SAR-TS: Currently, the performance of SAR-TS does not significantly exceed other methods.
        - For Masked-AR: The comparison between FMT and Masked AR is incomplete, as no baseline methods use a comparable parameter size to FMT-L or FMT-B. From the results presented, the improvements brought by SAR are also limited (e.g., a 0.05 lower FID than MaskGIT while having 50% more model parameters). Additionally, strong baselines like MAR-L and MAR-B from [2] are not not included.
        - For VAR: There is also no valid comparison with VAR.
        - For LlamaGen: While FMT performs better than the LlamaGen baseline, I would like to confirm what modifications are made by SAR compared to the LlamaGen baseline aside from the new architecture. It seems the token number per set is 1, and Tab.5 shows that raster order performs the best, both of which are settings used by LlamaGen.
    - Additionally, there is a certain dependency between various choices in training and inference, making it difficult to train a model that supports different steps and orders in inference while achieving optimal results. For example, from Tab.5, the negative impact of order mismatch appears significant; From Fig.8 and related discussions in the paper, the mismatch in steps can lead to undesirable results. Therefore, the flexibility of SAR seems somewhat limited.

    Given these concerns, although SAR is a new framework offering various options, the results do not show superiority over existing methods, nor is it fully flexible. The pratical advantages of SAR needs to be further clarfied.

3. Some statements need further clarification:
    - Since random-2-random appears to be a general form of training, which supports predicting an arbitrary number of tokens based on an arbitrary number of tokens, the "mismatch" between MAR's training and inference in Tab.1 needs more explanation.
    - The definitions of $m_e$, $m_{ds}$, and $m_{dc}$ in the Alg.1 and Alg.2 seem to be missing (please correct me if I missed something, thank you).
    - In line 426, why does it state that only 64-sets can work for a few steps? From Fig.8, it seems that 4-sets perform best at 4-steps sampling?
    - As discuss, MAR is a special case of SAR with "random-2-random". However, the authors indicate the K of MAR as 1 in Tab.6 and Fig.2.
    - In the middle sub-figure of Fig.9, are the training steps and sampling aligned? Additionally, do "causal calculation" and "full calculation" refer to causal and full attention masks, respectively?

[1] Chang et al., MaskGIT: Masked Generative Image Transformer

[2] Li et al., Autoregressive Image Generation without Vector Quantization

[3] Tian et al., Visual Autoregressive Modeling: Scalable Image Generation via Next-Scale Prediction

**Questions:**

1. Why not use a decoder-only architecture? At each step, the tokens sampled in the previous step and the positional embeddings of the next set of tokens are fed into the transformer as input, with other generated tokens as the prefix. Tihs apporach would also allow the use of KV cache to store the information of all previously generated tokens.
2. If the training and sampling steps are aligned for the middle sub-figure of Fig.9, it indicates that 2-step sampling achieves a FID of around 30, which is too good to be believed for me. Could the authors provide some code for verification?

---

> ### Author Response · Authors · 2024-11-19
>
> # Reply to Reviewer rmFX
>
> Thank you very much for your careful reading and valuable feedback! Below we will address the concerns:
>
> ## 1. Further Clarification on Differences between SAR and other Methods
>
> i) Compared to classical AR, a) SAR can be faster at inference, and b) SAR enables image editing, such as inpainting/outpainting due to the flexibility of order.
>
> ii) Compared to MAR, a) the inference speed is three times faster (can be referred to 5). b) SAR offers a more flexible way for causal learning. During causal training, SAR can simply incorporate any modality just by setting or adjusting the generalized causal mask. For example in video generation, we may only modify the mask to reflect the time order, and it can be trained as a video generator. Moreover, due to the causal manner of language, SAR can also easily facilitate multimodal learning.
>
> iii) Compared to VAR, a) SAR directly operates with common VAE, while VAR needs a customized multiscale VAE. b) SAR is more flexible in processing images, videos and other modality or multimodality.
>
> ## 2. Further Discussion on Experiments
> i) When compared with LlamaGen, all settings are the same except the model.
>
> ii) Since SAR-TS is new to AR models, we mainly focus on evaluating SAR-TS while do not explore much of other settings.
>
> iii) On the performance of SAR-TS. Actually, we find that the image quality of SAR-TS is good, but there are serious framing misalignment issues. For example, the randomly generated 8 images in Figure 15 in the Appendix. Among large number of generated images, the subjects frequently encounter problems of extremely large, extremely small, or only part displayed. We think it leads to the suboptimal FID.  Further, the issue may come from the strong reliance of SAR-TS on position embedding. Classical AR methods rely on static relative position relation, i.e., the next-token relation, so AR does not rely much on position embedding. But SAR-TS uses random order, it perceives position completely from position embedding. It also explains that SAR-AR gets better results than the AR baseline, but falls behind when using random orders. We think better position strategy may solve this problem. And for further verifying the generation ability of SAR-TS, we also train a text-to-image model, which can generate high-quality and high-resolution images after very short training (200k iterations with batchsize of 256).
>
> ## 3. Further Clarification of Statements
>
> i) The mismatch of MAR training. During training of MAR, it always randomly masks 70%-100% tokens; but at inference, the generated (unmasked) tokens grow gradually. For example, there are 70% unmasked tokens and the model is to predict the rest 30%. This scenario is not seen during training.
>
> ii) Sorry for that, and m_e, m_{ds}, m_{dc} mean causal masks for encoder self-attention, decoder self-attention and decoder cross-attention.
>
> iii) We think apart from 64-set and 256-set training, other variants lose too much performance. So we said that only 64-set training is suitable for few-step generation.
>
> iv) The '2' in 'random-2-random' means 2 sets. During training, MAR randomly divides tokens into 2 sets with random order, but it only calculates loss on the masked part. And we use K to denote the number of output sets that is used in calculating loss. So K=1.
>
> v) In the middle of Figure 9, the sampling steps are always 64. Sorry for misleading. It is true that "causal calculation" and "full calculation" refer to causal and full attention masks, respectively.
>
> ## 4. Questions
>
> i) If using such model, in the attention calculation, the seen part conducts causal attention, and the position embedding part conducts self- and position-to-seen causal attention. It allows KV cache, but does not allow parallel causal training. Based on this model, we turn to use whole position embeddings and a encoder-decoder architecture. Your comments also coincide with our idea. By the way, it also works with only the decoder of FMT, but the performance may degrade.
>
> ii) The sampling steps are 64 as said in 3 (v).

---

> ### Author Response · Authors · 2024-11-19
>
> ## 5. Supplementary on Training/Inference Speed Gain from KV Cache
>
> For a comprehensive comparison, we examine the time cost of FMT-XL (for text-to-image generation) when generating one 1024x1024 image using one A100 GPU. The generation order of AR is raster, while those of MAR and SAR are both random (actually the order does not affect speed). The results are as follows:
> | Setting | KV cache | 64 steps | 128 steps | 4096 steps |
> |---------|----------|-----------|------------|-------------|
> | AR | ✓ | - | - | 174.49s |
> | MAR | ✗ | 9.66s | 19.22s | 685.77s |
> | SAR-TS | ✗ | 7.45s | 14.72s | 606.35s |
> | SAR-TS | ✓ | 2.82s | 5.78s | 174.49s |
>
> We empirically find that there is no visual difference between using 64 steps and more. And, the visualization in Figure 1 and Figure 13 (in the Appendix) are all sampled with **64** steps. We can see that the inference speed of SAR is > 60 times than that of AR with KV cache enabled. Notably, in the transformer decoder, MAR applies global attention across all tokens, while the number of tokens processed in AR and SAR-TS increases gradually. Consequently, even without KV cache, the inference time for SAR-TS is shorter than that of MAR; when KV cache is enabled, SAR-TS is three times faster than MAR with $64$ or $128$ steps.

---

> ### Comment · Reviewer_rmFX · 2024-11-29
> **Response to the Rebuttal**
>
> Thank you for the your response. I appreciate the comprehensive experiments presented in this work and the effort put into addressing my questions. However, I still have some concerns regarding the response:
>
> A1.
> (a) The authors claim that MAR is not suitable for learning other modalities. More explanation is needed to support this statement. In my understanding, as long as sequences and tokens are properly defined, MAR can also perform generation in the same way it does for images.
> (b) I mentioned VAR in my initial review because **it is a causal method that predicts a set of tokens in one step**, independent of how tokens and sequences are defined (e.g., based on resolution). The authors have not explained how this method differs from SAR.
>
> A2.
> While the authors have pointed out reasons for the suboptimal results, they have not addressed the problem itself. The practical value of SAR-TS remains unclear to me.
>
> A3.
> Thank you for the clarification.
>
> A4.
> Thank you for the clarification.
>
> A5.
> The comparison of time between MAR and SAR seems unfair, as the authors do not show the performance of the two methods.
>
> In Summary, the authors' response does not address my primary concerns: the insights provided by SAR are limited, and the results are not compelling enough.

---

### Official Review · Reviewer_vUCC · 2024-11-03

**Soundness:** 3
**Presentation:** 3
**Contribution:** 3
**Rating:** 5
**Confidence:** 4

**Summary:**

This paper introduces Set AutoRegressive Modeling (SAR), a new paradigm for autoregressive image generation that generalizes conventional autoregressive (AR) models by allowing the sequence to be split into arbitrary sets containing multiple tokens, rather than outputting each token in a fixed order. The authors propose a Fully Masked Transformer (FMT) architecture to implement SAR. The paper shows that existing AR variants (like AR and Masked AR) are special cases within the SAR framework, and SAR enables smooth transitions between them.

**Strengths:**

1. Authors present a unified framework that generalizes existing AR approaches.

2. Good T2I result demonstrates real-world applicability through text-to-image generation.

**Weaknesses:**

1. The performance of SAR-TS is suboptimal compared to pure AR or MAR approaches.

2. The authors claim that  Fully Masked Transformer is the first work supporting KV cache for AR image generation. llamagen already has support this feature as llamagen is a causal transformer.

**Questions:**

The paper's presentation is mainly based on the discrete tokenizer. Does it support the continuous tokenizer AR method, like MAR. Could the authors detail the (SAR,K=1) results below mask AR in Table 4.

---

> ### Author Response · Authors · 2024-11-19
>
> # Reply to Reviewer vUCC
>
> We thank the reviewer for the feedback. And we will address the concerns as follows:
>
> ## 1. Possible Reason for Suboptimal FID of SAR-TS
>
> Based on observation, we find that the image quality of SAR-TS is good, but there are serious framing misalignment issues. For example, the randomly generated 8 images in Figure 15 in the Appendix. Among large number of generated images, we find that subjects frequently encounter problems of extremely large, extremely small, or only part displayed. We think this leads to the suboptimal FID.  Further, the framing misalignment issue may come from the strong reliance of SAR-TS on position embedding. Classical AR methods rely on static relative position relation, i.e., the next-token relation, so AR does not need position embedding. But SAR-TS uses random order, the perception of position completely comes from position embedding. It also explains that SAR-AR gets better results than AR baseline, but falls behind when using random orders.
>
> ## 2. The Claim of Fully Masked Transformer
>
> Our claim about FMT is, it enables causal learning with any sequence order and any output intervals, which means: i) the training is causal and parallel, ii) at inference, it can output tokens in any order, and can output multiple tokens at once. LlamaGen only satisfies i), but not ii).
>
> ## 3. Questions
>
> SAR is a paradigm on training/inference, and it is not limited to cross-entropy loss. With diffusion loss, it will work like MAR. K=1 in SAR means that the loss is calculated with only one group of output tokens at each training iteration, as shown in Figure 2. With causal mask disabled, conceptually it is the same as MAR (but the model and loss are different).

---

### Official Review · Reviewer_7W6R · 2024-11-05

**Soundness:** 3
**Presentation:** 2
**Contribution:** 2
**Rating:** 5
**Confidence:** 3

**Summary:**

This paper introduces a Set AutoReggressive Modeling (SAR) as a new paradigm for AutoRegressive image generation. By definition, SAR is a much more flexible way compared with AR which generate tokens in a fixed raster order, and covers several variants including Masked AR, as well as VAR. To support the use of SAR, the paper introduces a new variants of the original Transformer architecture, named as fully masked Transformer. Experiments on ImageNet and text-to-image generation has shown that SAR outperforms previous benchmarks including LlamaGen, MaskGiT under comparable settings, which proves the effectiveness of proposed method.

**Strengths:**

- The exploration of set AR could be helpful for future works that uses AR models for image/video generation.

- The paper provides a good way to transform traditional Transformers architecture into fully masked transformer.

- This paper conducts quite a lot experiments and ablations that tests different variants of the model (which I appreciate).

**Weaknesses:**

- The idea of using cosine schedule to determine the number of tokens seems not novel [1, 2]

- Although the paper includes many different ablations and variants of SAR design, most of them are probably not very necessary. For example, the ablation of sequence order settings shows raster is the best (Fig. 5), and training/inference with different order settings shows aligned training/inference works the best (Table 5). Both of these experiments are quite intuitive (without any surprise).

[1] Xie, Jinheng, et al. Show-o: One single transformer to unify multimodal understanding and generation.

[2] Huiwen Chang, Han Zhang, Lu Jiang, Ce Liu, and William T Freeman. Maskgit: Masked generative image transformer.


Writing typos:
- line 231-232: Eq. equation 2/3

**Questions:**

NA

---

> ### Author Response · Authors · 2024-11-19
>
> # Reply to Reviewer 7W6R
>
> We appreciate the reviewer for the helpful comments. Below, we address each of the points raised:
>
> ## 1. Using Cosine Schedule
>
> We use cosine schedule for its simplicity in performance evaluation, and it is not our contribution.
>
> ## 2. Questions on Ablation study
>
> Since FMT uses position embedding to indicate positions, it treats any sequence order equally. As a result, we cannot judge which order performs best in advance. For the results on aligned order, we will put them into the Appendix. Thanks for your advice.

---

> > ### Comment · Reviewer_7W6R · 2024-11-23
> >
> > Thanks for the authors' response. I truly appreciate the thorough experiments conducted in this work. However, I remain slightly concerned about the overall insights this paper offers for future research. Therefore, I have decided to maintain my score. Thank you!

---

### Official Review · Reviewer_QJJo · 2024-11-05

**Soundness:** 4
**Presentation:** 2
**Contribution:** 2
**Rating:** 5
**Confidence:** 4

**Summary:**

This work investigate image generation with discrete tokens. It proposes Set AutoRegressive Modeling (SAR), which can generate image tokens in arbitrary order and intervals. The authors also develop a variant of encoder-decoder Transformer named FMT to enable KV cache. Experiments on ImageNet show competitive performance and the model also shows capability in text2image generation.

**Strengths:**

1. The paper investigates an interesting problem to make autoregressive image generation model more efficient.
2. The work includes standard benchmarks like ImageNet 256 and achieves competitive performance.

**Weaknesses:**

1. Though the work is motivated by a more flexible framework for image generation, the best performing setting is still using raster order as standard autoregressive model (table 5).
2. It misses some comparison to baseline models to help understand the benefit of the proposed method.

More details in Questions section.

**Questions:**

1. How much training/inference speed gain does KV cache give? Some benchmark with AR/MAR models would be helpful to understand the benefits of the model.
2. What does SAR-TS mean? I may miss something, but I'm confused about how this transition state variant is built.
3. To better show the benefit of the proposed method, it would be valuable to compare SAR with MAR with the same fixed order both on FID and inference speed.
4. The model shows nice visualization on text2image generation. Are there any metrics to evaluate the performance over different inference settings like CLIP score?

---

> ### Author Response · Authors · 2024-11-19
>
> # Reply to Reviewer QJJo
>
> We appreciate the reviewer's feedback and questions regarding our work. Below, we address each of the points raised:
>
> ## 1. The Meaning of SAR-TS
>
> "TS" denotes transition state, i.e., the variant that can output tokens in **any order** with **few-step** inference. The definition is in the caption of Table 4.
>
> ## 2. Training/Inference Speed Gain from KV Cache
>
> First, we would like to explain that the visualization in Figure 1 and Figure 13 (in the appendix) are all sampled with **64** steps (the total number of tokens is **4096**). And we find that there is no visual difference between using 64 steps and more. For a comprehensive comparison, we examine the time cost of FMT-XL (for text-to-image generation) when generating one 1024x1024 image using one A100 GPU. The generation order of AR is raster, while those of MAR and SAR are both random (actually the order does not affect speed). The results are as follows:
> | Setting | KV cache | 64 steps | 128 steps | 4096 steps |
> |---------|----------|-----------|------------|-------------|
> | AR | ✓ | - | - | 174.49s |
> | MAR | ✗ | 9.66s | 19.22s | 685.77s |
> | SAR-TS | ✗ | 7.45s | 14.72s | 606.35s |
> | SAR-TS | ✓ | 2.82s | 5.78s | 174.49s |
>
> We can see that the inference speed of SAR is > 60 times than that of AR with KV cache enabled. Notably, in the transformer decoder, MAR applies global attention across all tokens, while the number of tokens processed in AR and SAR-TS increases gradually. Consequently, even without KV cache, the inference time for SAR-TS is shorter than that of MAR; when KV cache is enabled, SAR-TS is three times faster than MAR with $64$ or $128$ steps.
>
> ## 3. Performance of Text-to-Image Generation
>
> Since we only train the T2I model under the setting of SAR trasition state, the performance is optimal with causal attention and random order. Therefore we cannot compare different inference settings (AR, MAR, SAR-TS) fairly. Besides, the AR inference time with full 4096 steps is quite long (3 min for one image), so it may not be practical to evaluate it at high resolution. Here, we test the performance with 64 steps and compare it with other existing methods/models. The results are as follows, where the test dataset is MJHQ-30K [1], results of other methods in the table is cited from SANA [2], and the FID metric and CLIP score is calculated using [3] and [4] respectively.
> | Model                            | Params (B)|  FID  $\downarrow$   | CLIP score $\uparrow$|
> |----------------------------------|----------|----------|------------|
> | SDXL [5]       |   2.6    | 6.63     | 29.03      |
> | PlayGroundv2.5 [1]|   2.6    | 6.09     | 29.13      |
> | SD3-medium [6]  |   2.0    | 11.92    | 27.83      |
> | FLUX-dev [7]            |   12.0   | 10.15    | 27.47      |
> | Ours                             |   0.9    | 14.25    | 31.23      |
> From the table we can see that the FID of ours is worse than other methods, which is aligned with the results on ImageNet-256, while its CLIP score is better. For limitted resources, our T2I model is only trained for 200k iterations with batchsize of 256, which can be one of the reasons for worse FID.
>
> [1] Li et al. Playground v2.5: Three Insights towards Enhancing Aesthetic Quality in Text-to-Image Generation
>
> [2] Xie et al. SANA: Efficient High-Resolution Image Synthesis with Linear Diffusion Transformers
>
> [3] https://github.com/mseitzer/pytorch-fid
>
> [4] https://github.com/Taited/clip-score
>
> [5] Podell et al. SDXL: Improving latent diffusion models for high-resolution image synthesis
>
> [6] Esser et al. Scaling rectified flow transformers for
> high-resolution image synthesis
>
> [7] Black Forest Labs. Flux, 2024. URL https://blackforestlabs.ai/

---

### Meta-Review · Area_Chair_33ef · 2024-12-19

**Metareview:**

This paper proposes an architecture for autoregressive modeling called a Fully Masked Transformer that predicts sets of tokens together, in an arbitrary order, as opposed to standard AR models that predict one token at a time in a fixed order. The architecture of FMT splits the standard decoder-only Transformer into an encoder that encodes image semantics and a decoder stage that decodes output positions and models interactions between semantics and output tokens. The method is flexible in that it allows varying smoothly between AR and MAR decoding. Experiments on class-conditional and text-image settings highlight the model’s capabilities including over multiple aspect ratios and image quality.

Strengths:
Paper focuses on efficiency and flexibility for high-resolution generative modeling, which is of interest to the community. Experiments are thorough, including ablations, and are applied on standard datasets and indicate the method is competitive.

Weaknesses:
The best performing model in the T2I setting is still the standard AR order model, calling into question the value of the proposed method at least in some cases. There is also some concern/confusion about limited novelty over existing methods.

Decision reasoning:
While the paper presents an interesting direction to unify autoregressive and masked modeling, and has a reasonably thorough experiment section, reviewers remain unconvinced about the practical value of the method and whether the proposed machinery leads to substantial improvements over existing technniques. I encourage the authors to revisit the reviews and try to improve the paper to better motivate the method, and potentially show where different orders are valuable.

**Additional Comments On Reviewer Discussion:**

Reviewers were uniformly skeptical about the supporting evidence in favor of the method, which suggests the authors need to put time into further revisions.

---

### Decision · Program_Chairs · 2025-01-22

Reject